# Knowledge, attitude, and practice towards COVID-19 and associated factors among students in Ethiopia: Systematic review and meta-analysis

**Tenagework Eseyneh Dagnaw**[1]*, **Amare Mebrat Delie**[1], **Tadele Derbew Kassie**[2], **Sileshi Berihun**[1], **Hiwot Tesfa**[1], **Amare Zewdie**[3]

1 Department of Public Health, College of Medicine and Health Sciences, Injibara University, Injibara, Ethiopia, 2 Department of Public Health, College of Medicine and Health Sciences, Debre Markos University, Debre Markos, Ethiopia, 3 Department of Public Health, College of Medicine and Health Science, Wolkite University, Wolkite, Ethiopia

* tenagneworke1@gmail.com

**Data Availability Statement:** All relevant data are within the paper and its Supporting Information files.

## Abstract

### Introduction

The World Health Organization (WHO) has not confirmed the eradication of the COVID-19 pandemic or the development of a cure. Ongoing research is necessary to update public understanding, attitudes, and behaviors. Therefore, this study aims to evaluate the knowledge, attitudes, and practices related to COVID-19 among students in Ethiopia.

### Objective

To estimate the pooled proportions and associated factors of knowledge, positive attitude, and prevention practice toward COVID-19 among students in Ethiopia.

### Methods

The study followed the Preferred Reporting Items for Systematic Review and Meta-Analysis (PRISMA) guidelines and was registered on the PROSPERO database. Data extraction was done using an Excel spreadsheet, and analysis was conducted using STATA. The quality of the articles was assessed using the Joanna Briggs Institute (JBI) grading approach. A narrative summary and random-effects model with a 95% confidence interval were used. Heterogeneity and publication bias were also assessed. The results were presented using a forest plot and tables.

### Result

Out of 2089 studies reviewed, only 16 met the inclusion criteria. The pooled proportions of COVID-19 good knowledge, positive attitude, and good prevention practice among students in Ethiopia were found to be 62.68% [95% CI 50.96–74.40, I2 = 98.93%], 60.22% [95% CI 51.64–68.80, I2 = 97.68%], 48.47% [95% CI, 44.16–52.78, I2 = 91.45%] respectively.

**Funding:** The author(s) received no specific funding for this work.

**Competing interests:** The authors have declared that no competing interests exist.

**Abbreviations:** AOR, Adjusted Odd Ratio; AAU, Addis Ababa University; BG, Benishangul-Gumuz; CI, = Confidence Interval; COVID-19, Corona Virus Disease 2019; JBI, Joanna Bridges Institute; KAP, Knowledge, Attitudes, and Practice; PRISMA, Preferred Reporting Items for Systematic Review and Meta-Analysis; SNNPR, Southern Nations, Nationalities, Peoples Region; SRMA, = Systematic Review and Meta-Analysis; WHO, = World Health Organization.

Factors that affected knowledge included marital status, family size, sex, study year, and residency. Knowledge and residency were identified as predictors of attitude. Determinants of practice included knowledge, attitude, sex, study program, and residency.

## Conclusion

The pooled proportion of students in Ethiopia with good knowledge, positive attitudes, and good practices regarding COVID-19 was low. Policymakers, the Ministry of Health, Public Health Institutions, and other stakeholders should intensify their efforts on COVID-19 and develop interventions aimed at females, rural residents, and extension students. The protocol was registered on the PROSPERO database with ID = CRD42023478234.

## Introduction

The SARS-cov-2 virus is the infectious agent that causes coronavirus disease 2019 (COVID-19) [1]. The World Health Organization (WHO) classified the COVID-19 outbreak as a global health emergency on January 30, 2020 [2], and as a pandemic of COVID-19 on March 11, 2020 [3]. The first COVID-19 case in Africa was reported in Egypt on February 14, 2020 [4]. A 48-year-old Japanese male tested positive for COVID-19 in Addis Ababa, Ethiopia, after traveling from Japan to Burkina Faso before arriving in Ethiopia [5].

COVID-19 affects individuals differently, with most experiencing mild to severe illness and recovering without hospitalization. Serious disease is more common in the elderly and those with underlying medical issues [1].

Frequent symptoms include fever, coughing, exhaustion, loss of taste or smell, confusion, chest pain, and confusion, while severe symptoms include breathing difficulties, speech loss, mobility loss, and dementia [1]. COVID-19 is primarily transmitted through small droplets from the nose or mouth, and anyone infected can spread it, even without symptoms [6].

To prevent and slow down transmission, maintain a distance of at least 1 meter, wear a mask, frequently wash hands, get vaccinations, and follow local guidance [1]. The success of illness prevention efforts is significantly influenced by the client's behavior [7]. The management approach must be followed by all to stop the spread of COVID-19, which is influenced by people's knowledge, attitudes, and practices (KAP) regarding the virus [8].

COVID-19 has significantly impacted various aspects of the population, necessitating ongoing assessments of knowledge, attitudes, and practices to prevent disease transmission. Different populations have varying knowledge, attitudes, and prevention practices for COVID-19, necessitating tailored interventions, health education, and policies for effective disease prevention and management.

Studies have addressed knowledge, attitudes, and practices toward COVID-19 among different populations in Ethiopia. There are also meta-analyses among the general population [9], chronic patients [10], health professionals [11], and pregnant women [12] but no study assessed students' COVID-19 knowledge, attitude, and practice in Ethiopia. Students' interaction during learning and campus life results in a higher risk of contracting infectious diseases.

This systematic review and meta-analysis (SRMA) will fill the above gap and help policymakers and stakeholders in designing prevention programs. It is also important for future researchers and scholars. This study aims to assess the pooled proportions and associated factors of knowledge, attitude, and practice towards COVID-19 among students in Ethiopia.

## Methods

### Study design and setting

This study was done to estimate the pooled proportion and predictors of knowledge, attitude, and practice towards COVID-19 among students in Ethiopia using the Preferred Reporting Items for Systematic Review and Meta-Analysis (PRISMA) [13]. (S1 Table).

### Search strategies

This study protocol was registered on the PROSPERO database to prevent duplication with ID = CRD42023478234 [14]. An amendment was made to the protocol by adding predictors as an additional outcome. Studies were searched until December 10, 2023. PubMed, Google Scholar, African Journals Online, Research for Life, and Science Direct were the databases used to identify articles. A manual search was also done to get unpublished research from various repositories. Different keywords were used: COVID-19, Knowledge, attitude, practice, Ethiopia, students, associated factor, determinant, predictors, and mesh terms using Boolean operators. (S2 Table).

### Eligibility criteria

All published and grey literature, freely accessible full-text human studies done in Ethiopia which includes COVID-19 knowledge or attitude or practice, and/or associated factors were included. The articles incorporated in this review were exclusively written in English and pertained to cross-sectional study designs, with a cutoff date of December 10, 2023. Studies that scored less than five on the quality assessment criteria using the criteria of JBI for observational studies were excluded from this study.

### Outcomes measures

**Main outcome.**   Knowledge, attitude, and practice towards COVID-19 among students in Ethiopia. Each outcome was described using proportion.

**Additional outcome.**   Determinants of Knowledge, attitude, and practice towards COVID-19 among students in Ethiopia. Each outcome was described using AOR.

### Operational definitions

**Knowledge.**   Stored information about COVID-19 in the participant's mind. A score was given for each participant and the score was summed. A score greater than or equal to the mean/median is called good knowledge.

**Attitude.**   Feeling towards the COVID-19 prevention practice. A score was given for each participant and the score was summed. A score greater than or equal to the mean/median is called a positive attitude.

**Practice.**   Activity or action to prevent COVID-19. A score was given for each participant and the score was summed. A score greater than or equal to the mean/median is called good practice.

**Student.**   A person who is learning at schools, colleges, and universities.

### Data extraction

Any duplicate studies were removed once all research had been integrated into the Zotero reference management system. Three authors (TED, TDK and AZ) examined each article's title, abstract, objectives, and whole content thoroughly. The articles that had been identified as

possibly relevant were assessed further by going over the entire document, beginning with the titles, goals, methodology, subjects, and conclusions. The article was evaluated using the checklist. During the disagreement, we allowed additional personal judgment to reach a decision based on the same criteria.

All necessary data from each included article was extracted using a predefined data extraction format by three authors (TED, HT and AMD). Data on the extent of knowledge, attitude, and practice and associated factors towards COVID-19 among Ethiopian students was collected. The necessary information was collected using an Excel spreadsheet.

### Risk of bias (quality) assessment

The included papers' quality was evaluated using the Joanna Bridges Institute (JBI) grading approach for observational studies to reduce the risk of biases [15]. The authors (AMD, AZ and SB) examined the quality of the study separately and finally decided together. During the disagreement, a discussion was done and the disagreement was solved. (S3 Table).

### Data processing and analysis

Data was exported from a Microsoft Excel spreadsheet into STATA version 17 for additional analysis. Qualitative synthesis was applied as a narrative summary. A random-effects model at a 95% confidence interval was performed to estimate the outcomes. $I^2$ statistics were used to estimate heterogeneity among studies and the results of this test were interpreted conservatively. Subgroup analysis was done based on the study period, population type, and data collection methods.

Egger's test, Begg correlation test, and funnel plots were performed to assess publication bias. The P-value>0.05 indicated that there was no publication bias. The expected average proportion and AOR estimate of all possible studies were presented using confidence intervals. A forest plot and tables were used to present the result.

## Results

### Characteristics of included studies

From different databases and repositories, studies were searched and a total of 2086 pieces of research were retrieved. Of the total of searched studies, 889 had been removed due to duplication. By reviewing the title and abstracts, 1124 and 52 studies had been removed, respectively. Twenty-one studies were reviewed for eligibility criteria, but only 18 studies fulfilled the inclusion criteria (Fig 1). Four articles did not fulfill the eligibility criteria because of low quality [16] and absence of outcome [10, 17]. The number of studies included for each outcome was different. The pooled proportions of knowledge, positive attitude, and prevention practice toward COVID-19 included 14, 13, and 11 studies respectively. All studies had a cross-sectional study design. (Table 1).

### Knowledge, attitude, and practice towards COVID-19 among students

The pooled proportions of COVID-19 knowledge, attitude, and practice among students in Ethiopia were 62.68% [95% CI: 50.96–74.40, $I^2$ = 98.93%], 60.22% [95% CI: 51.64–68.80, $I^2$ = 97.68%], 48.47% [95% CI: 44.16–52.78, $I^2$ = 91.45%] respectively (Figs 2–4).

**Publication bias.** The funnel plot was done for knowledge, attitude, and practice and seems asymmetrical for all (Figs 5–7). Statistical analysis (Egger test and Begg test) was performed to check for small study effects or publication bias. Both tests showed no small study effect or publication bias for COVID-19 practice. For COVID-19 knowledge, the Egger test

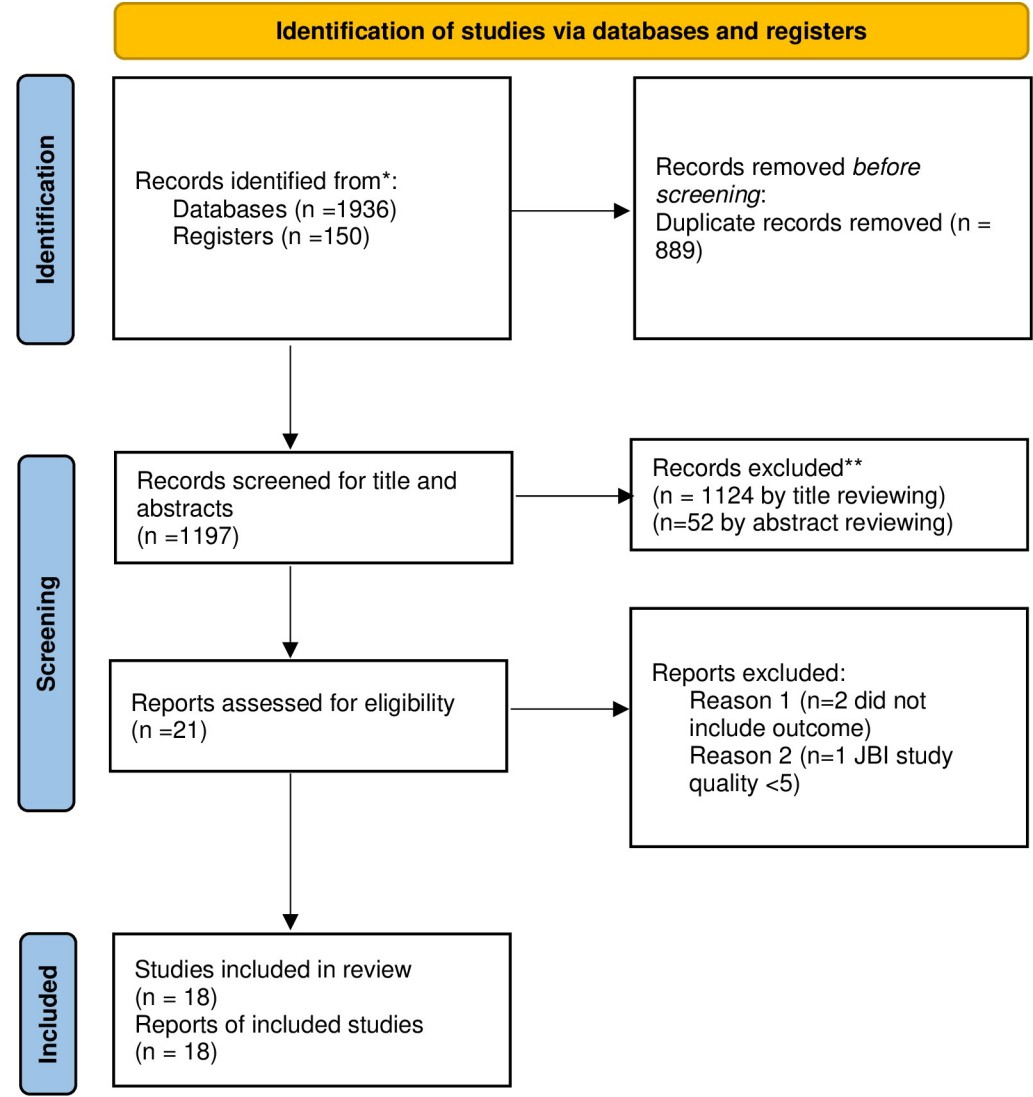

**Fig 1. Shows how included studies were selected using the PRISMA 2020 flow diagram.**

showed no small study effect or publication bias, even though the Begg test showed there was a small study effect or publication bias. Both tests confirmed that there was publication bias or a small study effect on COVID-19 attitudes among students in Ethiopia.

Publication bias cannot be removed, but we tried to manage it by doing a non-parametric trim and fill analysis of publication bias for knowledge and attitude. Trim and fill analysis results showed that no additional studies were needed and confirmed a similar pooled estimate for knowledge and attitude. Even if imputing studies on the right and left were done, there were no additional studies added.

Random-effects meta-regression was done for sample size, response rate, study period, population type, and data collection method. It indicated a problem with the sample size and population type for knowledge and attitude Meta-regression, respectively.

**Subgroup analysis by study period.** The pooled proportion of COVID-19 knowledge among students in 2020 was 57.42% [41.87–72.98], and gradually it became 71.08% [54.27–

**Table 1. Included studies characteristics for good knowledge, positive attitude, and good prevention practice among students in Ethiopia.**

| Authors, publication year | Type | Study period | Population | Setting | City, region | Sample size | Data collection method | Response rate (%) | JBI Score |
|---|---|---|---|---|---|---|---|---|---|
| Yesuf M and Abdu M (2022) [18] | Published | 2021 | Preparatory school students | Institutional | Mizan Tepi, SNNPR | 422 | Self-administered questionnaire | 96.20 | 7 |
| Tadesse AW et al (2020) [19] | Preprint | 2020 | Higher institutions students | Community based | Dessie, Amhara | 422 | Interviewer-administered questionnaire | 96.60 | 6 |
| Tazebew B et al (2023) [20] | Published | 2020 | BDU graduating students | Institutional | Bahir Dar, Amhara | 422 | Structured questionnaire | 96.40 | 7 |
| Getawa S et al (2022) [21] | Published | 2021 | Secondary school students | Institutional | Gonder, Amhara | 422 | Self-administered questionnaire | 93.60 | 8 |
| Berihun G et al (2021) [22] | Published | 2020 | Higher educational institution | Institutional | Amhara | 422 | Self-administered questionnaire | 96.40 | 8 |
| Mekonnen A (2021) [23] | AAU thesis | 2021 | Medical students in Addis Ababa | Institutional | Addis Ababa | 371 | Self-administered questionnaire | 94.10 | 8 |
| Aynalem YA et al (2021) [24] | Published | 2020 | Undergraduate students at Debre Berhan University | Institutional | Debrebrhan, Amhara | 634 | Self-administered questionnaire | 86.10 | 8 |
| Angelo AT et al (2020) [25] | Published | 2020 | Undergraduate students at Mizan Tepi University | Institutional | Mizan Tepi, SNNPR | 422 | Self-administered questionnaire | 95.30 | 7 |
| Walle TA and Temachu YZ (2023) [26] | Published | 2021 | Private college students | Institutional | Gonder, Amhara | 384 | Self-administered questionnaire | 100.00 | 9 |
| Feleke A et al (2022) [27] | Published | 2021 | High and preparatory school students | Institutional | Dessie, Amhara | 422 | Interviewer-administered questionnaire | 98.80 | 9 |
| Gutata D et al (2023) [28] | Preprint | 2021 | College students | Institutional | Assosa, BG | 535 | Interviewer-administered questionnaire | 97.00 | 9 |
| Handebo S et al (2021) [29] | Published | 2020 | High and preparatory school students | Institutional | Gonder, Amhara | 403 | Self-administered questionnaire | 91.80 | 7 |
| Asefa L et al (2021) [30] | Preprint | 2020 | BHU undergraduate students | Institutional | Bule Hora, Oromia | 410 | Structured questionnaire | 93.40 | 5 |
| Tsegaw M et al (2022) [31] | Published | 2021 | UG regular undergraduate students | Institutional | Gonder, Amhara | 416 | Self-administered questionnaire | 97.40 | 9 |
| Wogayehu B et al (2020) [32] | Preprint | 2020 | Final year students at Arbaminchi Health Science College | Institutional | Arbaminchi, SNNPR | 304 | Interviewer administered questionnaire | 100.00 | 9 |
| Feleke A et al (2022) [33] | Published | 2021 | High and preparatory school students | Institutional | Dessie, Amhara | 422 | Self-administered questionnaire | 98.80 | 9 |
| Tadese M et al (2022) [34] | Published | 2020 | Regular Debre Berhan University students | Institutional | Debrebrhan, Amhara | 713 | Self-administered questionnaire | 95.80 | 9 |
| Tadese M et al (2021) [35] | Published | 2020 | Regular Debre Berhan University students | Institutional | Debrebrhan, Amhara | 713 | Self-administered questionnaire | 95.80 | 9 |

87.90] in 2021. The subgroup analysis for the COVID-19 attitude among students indicated 61.52% [56.56–66.49] in 2020, even if the result showed a slight reduction in 2021, 58.70% [39.64–77.76]. When a comparison was made on the proportion of COVID-19 pooled prevention practice based on the study period, the proportion dropped from 50.53% [43.63–57.43] to 46.42% [41.16–51.68] in 2020 and 2021, respectively. (Table 2).

**Subgroup analysis by population type.** The subgroup analysis based on population type in the study revealed that among all higher institution students, including college and university students, the highest proportion of COVID-19 knowledge was 69.60%, while the lowest was among medical students in higher institutions. High school students, college students, and

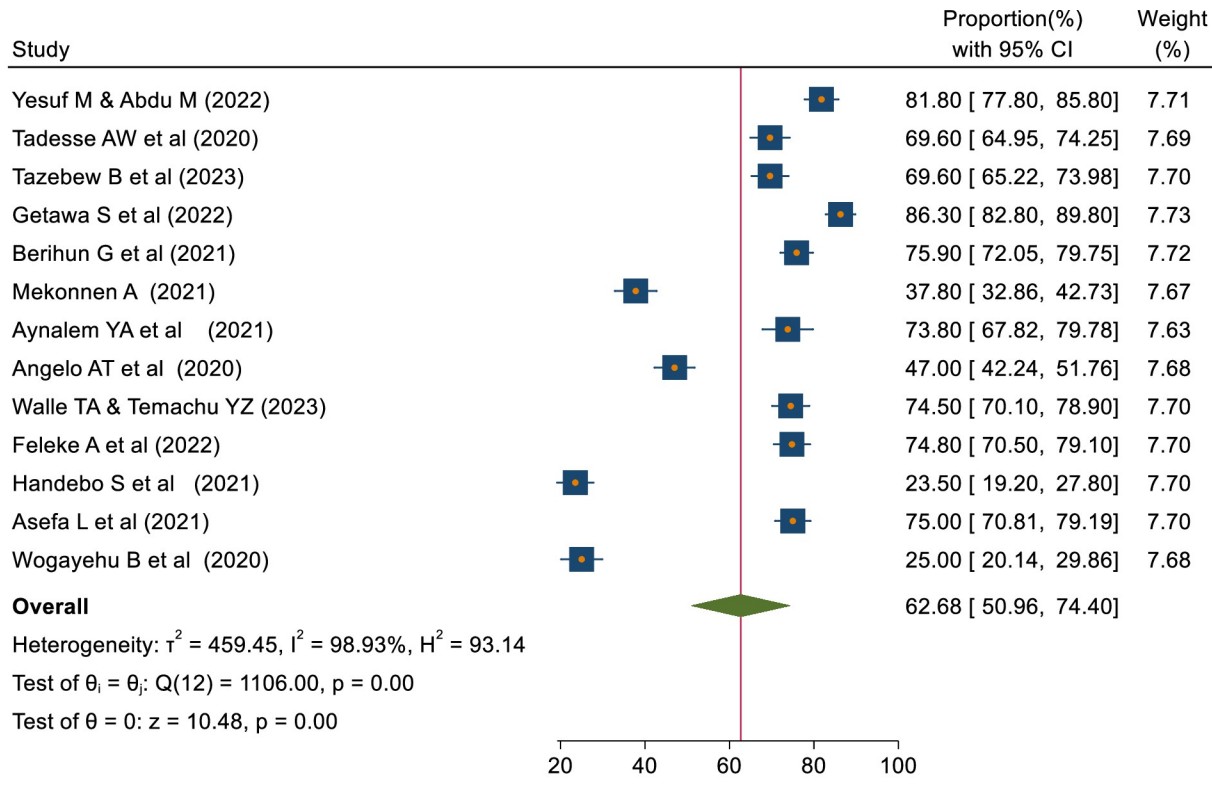

Fig 2. Forest plot showing COVID-19 knowledge among students in Ethiopia.

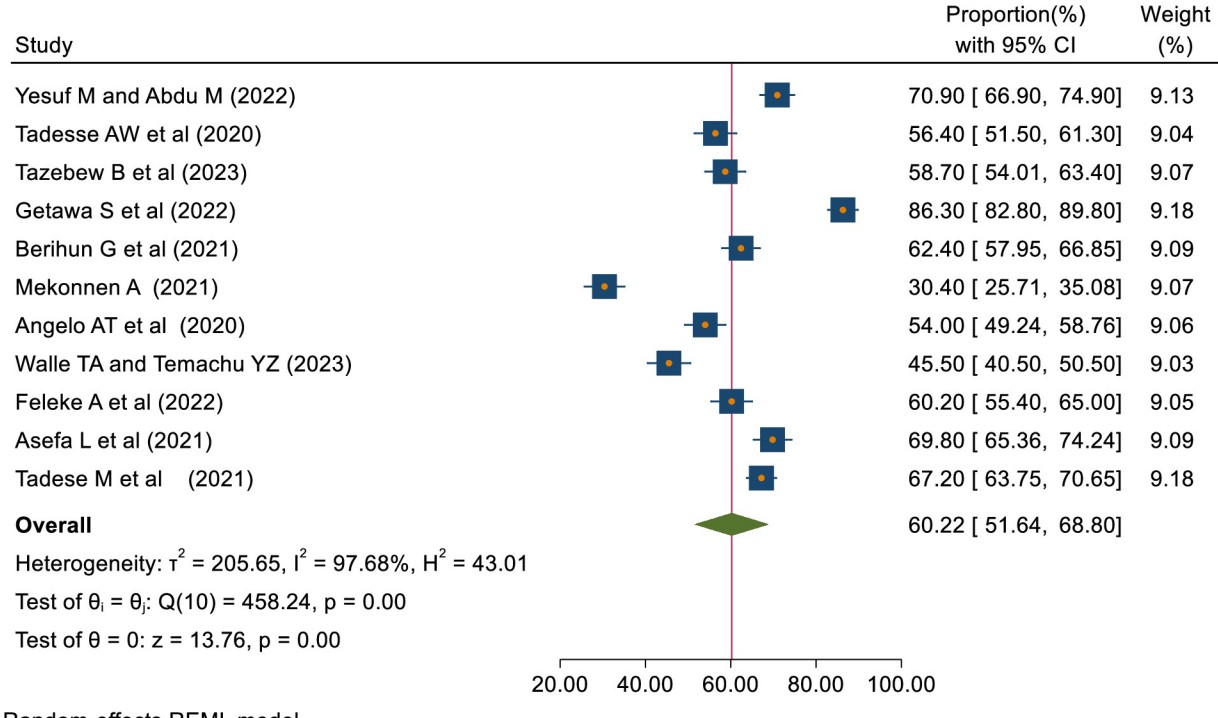

Fig 3. Forest plot showing COVID-19 positive attitude of students in Ethiopia.

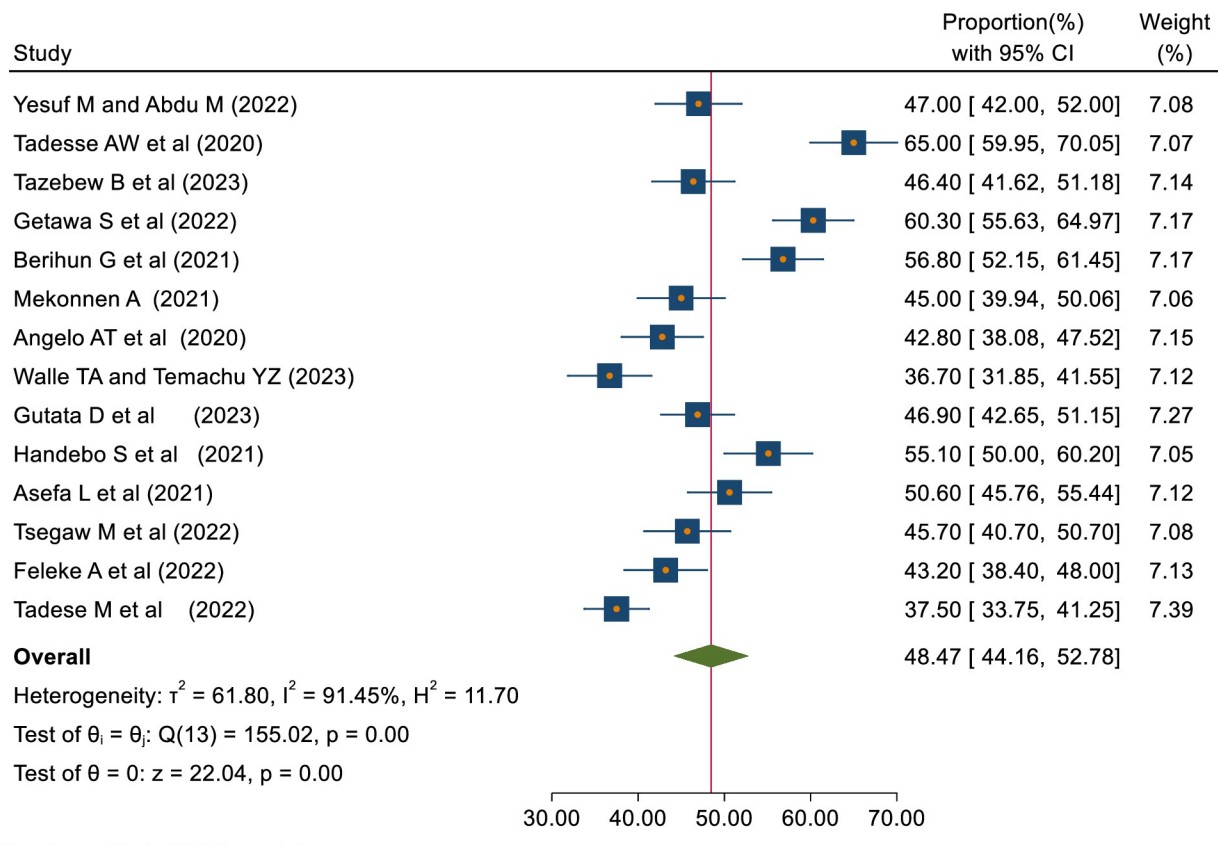

**Fig 4. Forest plot showing COVID-19 prevention practice among students in Ethiopia.**

university students had pooled proportions of knowledge at 66.61%, 49.76%, and 68.27%, respectively. Regarding attitudes, high school students had the highest proportion at 72.54%, and medical students had the lowest at 30.40%. The pooled proportions of COVID-19 prevention practice were 36.70% for college students, 45% for medical students, 46.6% for university students, 51.41% for high school students, and 65% for students of all higher institutions. (Table 2).

**Subgroup analysis by data collection method.** The sub-group analysis based on the data collection method showed the proportion of COVID-19 knowledge was 72.34%, 62.59%, and 56.48% among studies that used structured questionnaires, self-administered questionnaires, and interviewer-administered questionnaires, respectively. Around fifty-eight percent, 59.58%, and 64.25% pooled proportions of attitude were among studies that used interviewer-administered questionnaires, self-administered questionnaires, and structured questionnaires, respectively. In the case of COVID-19 prevention practice, the lowest and highest pooled proportions were 46.98% and 55.9% among studies that used self-administered and interviewer-administered questionnaires, respectively. (Table 2).

**Sensitivity analysis.** A Leave-out-one sensitivity analysis was done for the proportion of good knowledge, positive attitude, and good practice to assess the effect of one study on the pooled effect. The result showed that no single study alone affects the pooled proportion of good knowledge, a positive attitude, and good practice. (Table 3).

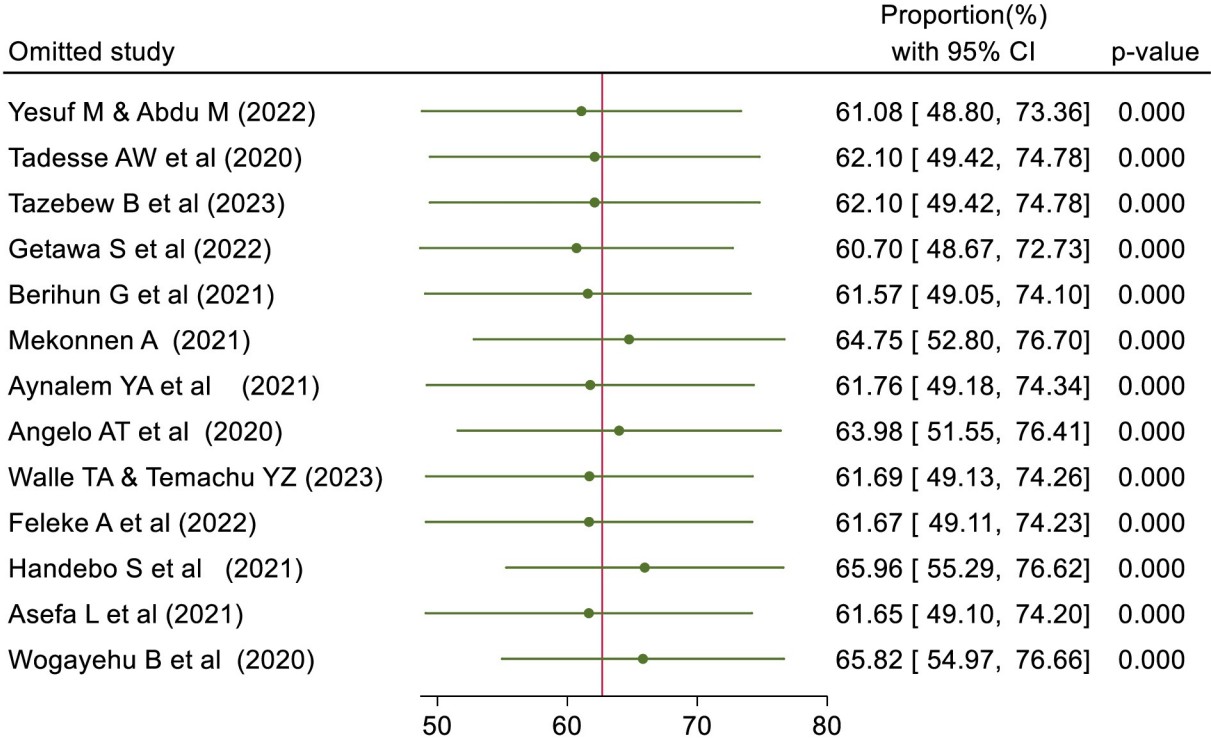

Fig 5. Funnel plot that shows the symmetry of COVID-19 knowledge studies among students in Ethiopia.

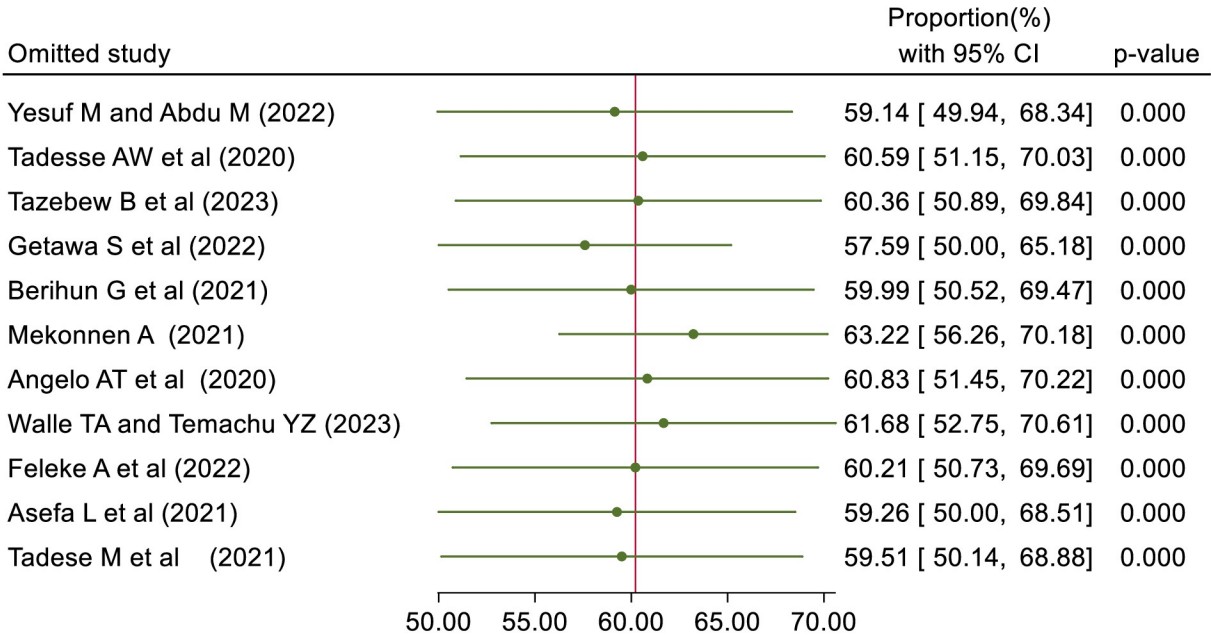

Fig 6. Funnel plot that shows the symmetry of COVID-19 positive attitude studies among students in Ethiopia.

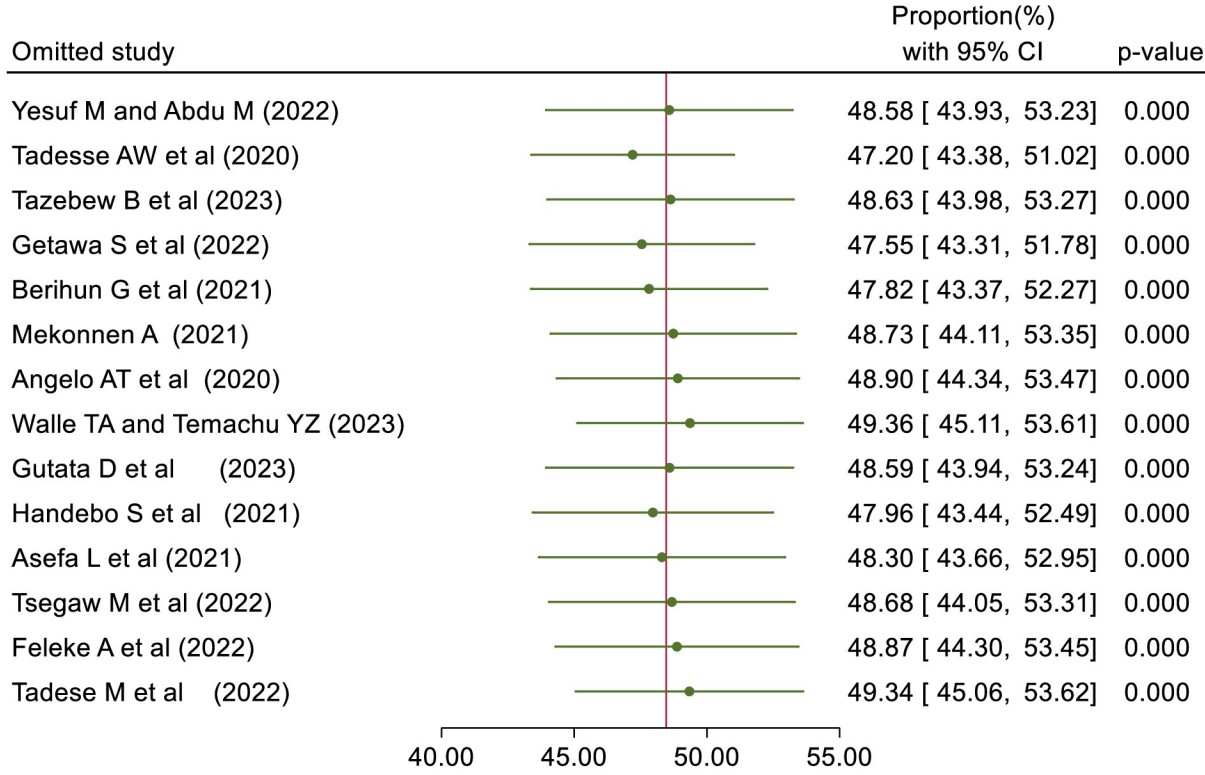

Random-effects REML model

**Fig 7. Funnel plot that shows the symmetry of COVID-19 prevention practice studies among students in Ethiopia.**

### Pooled associated factors of knowledge towards COVID-19 among students in Ethiopia

Ethiopian students who are single are less likely to have good knowledge by 5% than married ones (AOR 0.95; 95% CI: 1.71–4.54). Comparing a family size greater than 5 to less or equal to 5, a former has 2.68 times more chance of having good knowledge than the latter at 95% CI [1.71–4.54]. Similarly, males are more likely to have good knowledge than females (AOR: 2.37; 95% CI: 1.23–4.55).

Students who live in urban have more good knowledge than rural students at AOR: 2.51 95% CI: [1.84–3.41]. Additionally, 4th-year students are more likely to have good knowledge than 2nd-year students by 2.77 times at 95% CI: [1.54–4.97]. (Table 4).

### Pooled associated factors of attitude towards COVID-19 among students in Ethiopia

Students who have good knowledge are 2.92 times more likely to have a positive attitude than their counterparts at 95% CI: [2.19–3.89]. Residency is one of the predictors of attitude in which rural dwellers are 33% less likely to have a positive attitude than urban dwellers [95% CI: 0.05–9.05]. (Table 5).

### Pooled associated factors of practice towards COVID-19 among students in Ethiopia

Students with good knowledge and positive attitudes are more likely to practice COVID-19 prevention methods than their counterparts (AOR: 3.87; 95% CI: 1.75–8.54) and (AOR: 3.34;

**Table 2. Subgroup analysis of good knowledge, positive attitude, and good prevention practice towards COVID-19 among students in Ethiopia.**

| Outcome | Variable | Subgroup | Total studies | Prevalence [95%CI] | I²(%) | P-value |
|---|---|---|---|---|---|---|
| **Good Knowledge** | Study period | 2020 | 8 | 57.42[41.87,72.98] | 98.94 | 0.00 |
| | | 2021 | 5 | 71.08[54.27,87.90] | 98.77 | 0.00 |
| | Data collection method | Self-administered questionnaire | 8 | 62.59[46.52,78.67] | 99.09 | 0.00 |
| | | Interviewer administered questionnaire | 3 | 56.48[25.51,87.45] | 99.26 | 0.00 |
| | | Structured questionnaire | 2 | 72.34[67.05,77.63] | 67.16 | 0.08 |
| | Population type | High school students | 4 | 66.61[38.8,95.14] | 99.51 | 0.00 |
| | | University students | 5 | 68.27[57.64,78.90] | 96.36 | 0.00 |
| | | College students | 2 | 49.76[1.25,98.27] | 99.54 | 0.00 |
| | | All higher institution students | 1 | 69.60[64.95,74.25] | 0 | 0 |
| | | Only medical students in higher institutions | 1 | 37.80 | 0 | 0 |
| **Positive Attitude** | Study period | 2020 | 6 | 61.52[56.56,66.49] | 86.94 | 0.00 |
| | | 2021 | 5 | 58.70[39.64,77.76] | 98.97 | 0.00 |
| | Data collection method | Self-administered questionnaire | 7 | 59.58[46.09,73.07] | 98.63 | 0.00 |
| | | Interviewer administered questionnaire | 2 | 58.33[54.61,62.06] | 15.18 | 0.28 |
| | | Structured questionnaire | 2 | 64.28[53.40,75.15] | 91.18 | 0.00 |
| | Population type | High school students | 3 | 72.54[57.69,87.39] | 97.50 | 0.00 |
| | | University students | 5 | 62.51[56.97,68.06] | 87.91 | 0.00 |
| | | College students | 1 | 45.50[40.50,50.50] | 0 | 0 |
| | | All higher institution students | 1 | 56.40[51.50,61.30] | 0 | 0 |
| | | Only medical students in higher institutions | 1 | 30.40[25.71,35.08] | 0 | 0 |
| **Good prevention Practice** | Study period | 2020 | 7 | 50.53[43.63,57.43] | 93.50 | 0.00 |
| | | 2021 | 7 | 46.42[41.16,51.68] | 88.17 | 0.00 |
| | Data collection method | Self-administered questionnaire | 10 | 46.98[42.01,5195] | 90.98 | 0.00 |
| | | Interviewer administered questionnaire | 2 | 55.90[38.16,73.63] | 96.54 | 0.00 |
| | | Structured questionnaire | 2 | 48.48[44.37,52.60] | 31.71 | 0.23 |
| | Population type | High school students | 4 | 51.41[43.80,59.02] | 89.70 | 0.00 |
| | | University students | 7 | 46.60[42.06,51.14] | 85.73 | 0.00 |
| | | College students | 1 | 36.70[31.85,41.55] | 0 | 0 |
| | | All higher institution students | 1 | 65.00[59.95,70.05] | 0 | 0 |
| | | Only medical students in higher institutions | 1 | 45.00[39.94,50.06] | 0 | 0 |

95% CI: 2.46–4.55) respectively. Regular students are 4.73 times more likely to have good practice towards COVID-19 than extension students at 95% CI: [0.01–1612.90]. Similarly, female students are 1.73 times more likely to have good COVID-19 prevention practices than male students [95% CI: 1.36–2.20]. In addition, rural students are less likely to have good practice than students in urban areas by 22% (AOR: 0.78; 95% CI: 0.31–20.00). (Table 6).

## Discussion

COVID-19 is a pandemic without a cure. Despite the efforts made by the Ethiopian government to prevent the spread of COVID-19, many people have suffered and died from it. The World Health Organization (WHO) has not confirmed the eradication of COVID-19 or the development of a specific treatment. Management of the virus focuses on alleviating symptoms rather than targeting the virus itself. Therefore, prevention has remained the most reliable approach to avoiding the disease since the beginning of the outbreak. Continuous research is crucial to implement interventions, advocacy, and health education and communication initiatives, especially to change the knowledge, attitudes, and practices related to COVID-19 among students in Ethiopia. For this reason, this systematic review and meta-analysis aimed to

**Table 3. Shows the leave-out-one sensitivity analysis result of knowledge, attitude, and practice.**

| Omitted study | Proportion (%) [Confidence interval (%)] | | |
| --- | --- | --- | --- |
| | Good Knowledge | Positive Attitude | Good Practice |
| Yesuf M & Abdu M (2022) | 61.08[48.80–73.36] | 59.14[49.94–68.34] | 48.58[43.93–53.23] |
| Tadesse AW et al (2020) | 62.10[49.42–74.78] | 60.59[51.15–70.03] | 47.20[43.38–51.02] |
| Tazebew B et al (2023) | 62.10[49.42–74.78] | 60.36[50.89–69.84] | 48.63[43.98–53.27] |
| Getawa S et al (2022) | 60.70[48.67–72.73] | 57.59[50.00–65.18] | 47.55[43.31–51.78] |
| Berihun G et al (2021) | 61.57[49.05–74.10] | 59.99[50.52–69.47] | 47.82[43.37–52.27] |
| Mekonnen A (2021) | 64.75[52.80–76.70] | 63.22[56.26–70.18] | 48.73[44.11–53.35] |
| Aynalem YA et al (2021) | 61.76[49.18–74.34] | NA | NA |
| Angelo AT et al (2020) | 63.98[51.55–76.41] | 60.83[51.45–70.22] | 48.90[44.34–53.47] |
| Walle TA & Temachu YZ (2023) | 61.69[49.13–74.26] | 61.68[52.75–70.61] | 49.36[45.11–53.61] |
| Gutata D et al (2023) | NA | NA | 48.59[43.94–53.24] |
| Feleke A et al (2022) | 61.67[49.11–74.23] | 60.21[50.73–69.69] | 48.87[44.30–53.45] |
| Handebo S et al (2021) | 65.96[55.29–76.62] | NA | 47.96[43.44–52.49] |
| Asefa L et al (2021) | 61.65[49.10–74.20] | 59.26[50.00–68.51] | 48.30[43.66–52.95] |
| Tsegaw M et al (2022) | NA | NA | 48.68[44.05–53.31] |
| Wogayehu B et al (2020) | 65.82[54.97–76.66] | NA | NA |
| Tadese M et al (2021) | NA | 59.51[50.14–68.88] | 49.34[45.06–53.62] |

Note: NA = the study was not included in the assessment of outcome pooled effect

assess the knowledge, attitudes, and practices regarding COVID-19 among students in Ethiopia.

The result of this study indicated that the pooled proportion of COVID-19 knowledge among students in Ethiopia was 62.68% [95% CI 50.96–74.40, $I^2$ = 98.93%]. This is in line with the study done among all populations [9] and pregnant women in Ethiopia [12]. In contrast, this is lower than the review done among health workers [11] and all populations in Ethiopia [36]. In addition, the result is lower than reviews done globally [37–39] and a review done among college students in China [40]. The reason can be differences in the study population, setting, and study period.

In this review, the pooled proportion of COVID-19 attitudes among students in Ethiopia was 60.22% [95% CI 51.64–68.80, I2 = 97.68%]. This is in line with the studies done among all populations [9, 36] and pregnant women in Ethiopia [12]. This is similar to the review done among health workers in Ethiopia [11]. On the contrary, this is lower than meta-analyses around the globe [37–39] and a review done among college students in China [40]. The discrepancy can be due to differences in study area, knowledge level, and accessibility of COVID-19 information.

This study revealed that the pooled proportion of COVID-19 practice among students in Ethiopia was 48.47% [95% CI, 44.16–52.78, I2 = 91.45%]. This result is consistent with the studies done among all populations [9, 36, 41], pregnant women [12], and chronic illness patients in Ethiopia [10]. However, this is lower than the review done among health workers in Ethiopia [11]. In addition, this is a lower result compared to meta-analyses done around the globe [37–39] and a review done among college students in China [40]. Poor knowledge and negative attitudes about COVID-19 among students can contribute to a lower level of prevention practice.

Single students are less likely to have good knowledge than married ones (AOR 0.95; 95% CI: 1.71–4.54). This might be due to communication and interaction among partners which increases the knowledge of a person towards COVID-19. This is also similar to findings from Bangladesh, in which married people are more likely to have good knowledge than unmarried

**Table 4. A meta-analysis of factors affecting good knowledge about COVID-19 among students in Ethiopia.**

| Study | AOR | Lower AOR | Upper AOR | Weight |
|---|---|---|---|---|
| **Marital status single vs. Married** | | | | |
| Tadesse AW et al (2020) | 2.30 | 1.02 | 5.19 | 49.59 |
| Wogayehu B et al (2020) | 0.40 | 0.19 | 0.85 | 50.41 |
| Pooled estimate | 0.95 | 0.17 | 5.29 | |
| Heterogeneity | $T^2 = 1.37$ | I2 (%) = 89.55 | $H^2 = 9.57$ | |
| **Family size >5 vs. 5 and less** | | | | |
| Tadesse AW et al (2020) | 2.27 | 1.47 | 3.50 | 56.10 |
| Feleke A et al (2022) | 3.30 | 1.99 | 5.46 | 43.90 |
| Pooled estimate | 2.68 | 1.86 | 3.85 | |
| Heterogeneity | $T^2 = 0.01$ | I2 (%) = 17.82 | $H^2 = 1.22$ | |
| **Sex male vs. Female** | | | | |
| Mekonnen A (2021) | 1.71 | 1.05 | 2.78 | 51.29 |
| Wogayehu B et al (2020) | 3.33 | 1.96 | 5.66 | 48.71 |
| Pooled estimate | 2.37 | 1.23 | 4.55 | |
| Heterogeneity | $T^2 = 0.16$ | I2 (%) = 69.94 | $H^2 = 3.33$ | |
| **Residency Urban vs. Rural** | | | | |
| Tazebew B et al (2023) | 2.27 | 1.46 | 3.53 | 48.78 |
| Getawa S et al (2022) | 5.60 | 1.77 | 17.71 | 7.17 |
| Aynalem YA et al (2021) | 4.30 | 1.74 | 10.60 | 11.68 |
| Wogayehu B et al (2020) | 2.00 | 1.16 | 3.44 | 32.37 |
| Pooled estimate | 2.51 | 1.84 | 3.41 | |
| Heterogeneity | $T^2 = 0.00$ | I2 (%) = 0.00 | $H^2 = 1.00$ | |
| **Study year 4th vs. 2nd** | | | | |
| Berihun G et al (2021) | 3.10 | 1.09 | 8.82 | 31.28 |
| Walle TA and Temachu YZ (2023) | 2.63 | 1.30 | 5.33 | 68.72 |
| Pooled estimate | 2.77 | 1.54 | 4.97 | |
| Heterogeneity | $T^2 = 0.00$ | I2 (%) = 0.00 | $H^2 = 1.00$ | |

people [42]. A study done among health professionals in Lima, Peru indicated that being married is a risk factor for low knowledge by six times [43].

A family size greater than 5 has 2.68 times more odds of having good knowledge than a family size of 5 and less (95% CI: 1.71–4.54). The reason might be due to communication and interaction towards COVID-19 among the members of the family.

**Table 5. A meta-analysis of factors affecting positive attitudes toward COVID-19 among students in Ethiopia.**

| Study | AOR | Lower AOR | Upper AOR | Weight |
|---|---|---|---|---|
| **Knowledge Good vs. Poor** | | | | |
| Tadesse AW et al (2020) | 3.23 | 2.08 | 4.99 | 43.43 |
| Berihun G et al (2021) | 3.00 | 1.59 | 5.66 | 20.52 |
| Feleke A et al (2022) | 2.55 | 1.58 | 4.12 | 36.05 |
| Pooled estimate | 2.92 | 2.19 | 3.89 | |
| Heterogeneity | $T^2 = 0.00$ | I2 (%) = 0.00 | $H^2 = 1.00$ | |
| **Residency Rural vs. Urban** | | | | |
| Tazebew B et al (2023) | 0.18 | 0.09 | 0.35 | 50.85 |
| Asefa L et al (2021) | 2.58 | 1.00 | 6.68 | 49.15 |
| Pooled estimate | 0.67 | 0.05 | 9.05 | |
| Heterogeneity | $T^2 = 3.37$ | I2 (%) = 95.05 | $H^2 = 20.51$ | |

**Table 6. A meta-analysis of factors affecting good prevention practices towards COVID-19 among students in Ethiopia.**

| Study | AOR | Lower AOR | Upper AOR | Weight |
|---|---|---|---|---|
| **Knowledge Good vs. Poor** | | | | |
| Yesuf M and Abdu M (2022) | 5.17 | 2.28 | 11.76 | 28.63 |
| Angelo AT et al (2020) | 1.89 | 1.23 | 2.91 | 36.85 |
| Gutata D et al (2023) | 6.51 | 3.76 | 11.25 | 34.51 |
| Pooled estimate | 3.87 | 1.75 | 8.54 | |
| Heterogeneity | $T^2 = 0.40$ | I2 (%) = 82.18 | $H^2 = 5.61$ | |
| **Attitude Positive vs. Negative** | | | | |
| Yesuf M and Abdu M (2022) | 4.30 | 2.35 | 7.87 | 14.41 |
| Tazebew B et al (2023) | 2.20 | 1.34 | 3.61 | 17.60 |
| Mekonnen A (2021) | 2.79 | 1.47 | 5.33 | 13.37 |
| Angelo AT et al (2020) | 2.68 | 1.74 | 4.10 | 19.88 |
| Gutata D et al (2023) | 6.51 | 3.76 | 11.25 | 15.98 |
| Feleke A et al (2022) | 3.33 | 2.10 | 5.28 | 18.76 |
| Pooled estimate | 3.34 | 2.46 | 4.55 | |
| Heterogeneity | $T^2 = 0.08$ | I2 (%) = 52.48 | $H^2 = 2.10$ | |
| **Residency Rural vs. Urban** | | | | |
| Tadesse AW et al (2020) | 0.35 | 0.23 | 0.53 | 20.37 |
| Tazebew B et al (2023) | 0.16 | 0.07 | 0.36 | 18.28 |
| Berihun G et al (2021) | 1.67 | 1.11 | 2.50 | 20.44 |
| Angelo AT et al (2020) | 1.74 | 1.14 | 2.66 | 20.36 |
| Tadese M et al (2022) | 1.56 | 1.07 | 2.28 | 20.54 |
| Pooled estimate | 0.78 | 0.31 | 20.00 | |
| Heterogeneity | $T^2 = 1.07$ | I2 (%) = 95.32 | $H^2 = 21.35$ | |
| **Program regular vs. Extension** | | | | |
| Tadesse AW et al (2020) | 0.26 | 0.12 | 0.55 | 51.25 |
| Tazebew B et al (2023) | 100.00 | 13.62 | 733.99 | 48.75 |
| Pooled estimate | 4.73 | 0.01 | 1612.90 | |
| Heterogeneity | $T^2 = 17.12$ | I2 (%) = 96.66 | $H^2 = 29.98$ | |
| **Sex female vs. Male** | | | | |
| Tsegaw M et al (2022) | 1.63 | 1.02 | 2.59 | 26.64 |
| Feleke A et al (2022) | 1.96 | 1.24 | 3.10 | 27.32 |
| Tadese M et al (2022) | 1.67 | 1.17 | 2.38 | 46.04 |
| Pooled estimate | 1.73 | 1.36 | 2.20 | |
| Heterogeneity | $T^2 = 0.00$ | I2 (%) = 0.00 | $H^2 = 1.00$ | |

The odds of having good knowledge are higher among males than females (AOR: 2.37; 95% CI: 1.23–4.55). This is also similar to findings from Bangladesh [42]. This may be because males have a high chance of getting information from friends and social media. In contrast, a study done among undergraduate medical students in Indonesia indicated that females have better knowledge than males [44].

Additionally, 4th-year students are more likely to have good knowledge than 2nd-year students (AOR: 2.77; 95% CI: 1.54–4.97. This may be due to knowledge storage increasing as a year of study increases. This is also supported by a study done among undergraduate medical students in Indonesia in which clinical students had more chance of good knowledge than preclinical [44].

Urban dwellers have more odds of good knowledge than rural dwellers at AOR: 2.51 95% CI: [1.84–3.41]. This is supported by a study done among adult Egyptians [45]. Similarly, rural

dwellers are less likely to have a positive attitude than urban dwellers (AOR: 0.67; 95% CI: 0.05–9.05). This is supported by a study among patients at the University Medical Center in Ho Chi Minh City, Vietnam [46]. The reason can be exposure to different health information, health education, and advocacy in urban areas. The odds of having a positive attitude are more likely among students with good knowledge than their counterparts at AOR: 2.92; 95% CI: [2.19–3.89]. This can be due to the nature of behavior in which knowledge precedes attitude; a person with enough COVID-19 knowledge will also have a positive feeling towards the prevention methods.

Students with good knowledge and positive attitudes are more likely to practice COVID-19 prevention measures than their counterparts (AOR: 3.87; 95% CI: 1.75–8.54) and (AOR: 3.34; 95% CI: 2.46–4.55) respectively. This may be because behavior has three domains; the cognitive, the affective and the psychomotor domain predispose for practice or the psychomotor domain. Similarly, a meta-analysis done in Ethiopia revealed that lack or poor knowledge [10, 41] and negative attitude are predictors of poor practice [41]. Meta-analysis in Ethiopia shows that good knowledge is associated with good practice [12] and a favorable attitude is associated with good practice [47].

Female students are more likely to have good COVID-19 prevention practices than males at AOR 1.73 [95% CI: 1.36–2.20]. This is supported by a study done in Ethiopia at Jimma town in which females had higher adherence to COVID-19 prevention than males [16]. This might be due to female ability and sensitivity to threat. In addition, the reason might be females' fear of threat may result in practicing the prevention measures. In contrast, findings from a meta-analysis done in Ethiopia showed that females are more likely to have poor practice than males [41]. SRMA, Global epidemiology of KAP on COVID-19, indicated that there is no significant difference between females and males at the KAP level [37].

The finding also indicated that regular students are more likely to have good COVID-19 practice than extension students (AOR: 4.73 95% CI: 0.01–1612.90). This may be due to different individual behaviors and risk perceptions of the students. In addition, rural students are less likely to have good COVID-19 prevention practices than students in urban areas (AOR: 0.78; 95% CI: 0.31–20.00). This is because rural people have low access to new information and cultural differences, more connections among people, and social networking. The result is supported by a meta-analysis done in Ethiopia [10, 12, 41, 47].

### Strength and limitation

This study is the first to estimate the pooled proportion of KAP towards COVID-19 and includes both published and unpublished articles among students in Ethiopia. The existence of non-standardized tools in the reviewed articles may represent a limitation, leading to variability that could affect the applicability of the results. Only freely accessible full-text articles written in English were included in this study, which may limit generalizability.

### Conclusions

In this SRMA, the pooled proportion of good knowledge, positive attitudes, and good prevention practices regarding COVID-19 among students in Ethiopia was low. The pooled factors affecting good knowledge are marital status, family size, sex, study year, and residency. The pooled factors affecting positive attitude are knowledge and residency. The pooled factors affecting good prevention practice are knowledge, attitude, sex, program (regular vs extension), and residency. Policymakers, the Ministry of Health, public health institutions, and other stakeholders should increase their work on COVID-19 and prepare interventions that target students. Future researchers should work on the impact of low KAP. The pooled

proportion of attitudes and practices that were eradicated in 2021 relative to 2020 underscores the urgent need for enhanced intervention, advocacy efforts, and policy reforms. Special intervention is needed among medical students due to the low proportion of positive attitudes. The result also pointed out the need to give attention to females, extension students, rural dwellers, and family size while preparing an intervention or health education and policy.

## Supporting information

**S1 Table. PRISMA 2020 checklist.**
(PDF)

**S2 Table. Data extraction strategies.**
(PDF)

**S3 Table. All studies identified and quality appraisal.**
(PDF)

## Acknowledgments

We acknowledge all authors of the included articles in this systematic review and meta-analysis.

## Author Contributions

**Conceptualization:** Tenagework Eseyneh Dagnaw.

**Data curation:** Tenagework Eseyneh Dagnaw, Tadele Derbew Kassie, Sileshi Berihun, Amare Zewdie.

**Formal analysis:** Tenagework Eseyneh Dagnaw, Amare Mebrat Delie, Hiwot Tesfa, Amare Zewdie.

**Funding acquisition:** Tenagework Eseyneh Dagnaw, Amare Mebrat Delie, Tadele Derbew Kassie, Sileshi Berihun, Hiwot Tesfa, Amare Zewdie.

**Investigation:** Tenagework Eseyneh Dagnaw, Amare Mebrat Delie, Hiwot Tesfa, Amare Zewdie.

**Methodology:** Tenagework Eseyneh Dagnaw, Tadele Derbew Kassie, Sileshi Berihun, Amare Zewdie.

**Project administration:** Tenagework Eseyneh Dagnaw, Tadele Derbew Kassie, Sileshi Berihun, Amare Zewdie.

**Resources:** Tenagework Eseyneh Dagnaw, Amare Mebrat Delie, Tadele Derbew Kassie, Sileshi Berihun, Hiwot Tesfa, Amare Zewdie.

**Software:** Tenagework Eseyneh Dagnaw, Amare Mebrat Delie, Tadele Derbew Kassie, Sileshi Berihun, Hiwot Tesfa, Amare Zewdie.

**Supervision:** Tenagework Eseyneh Dagnaw, Amare Mebrat Delie, Hiwot Tesfa, Amare Zewdie.

**Validation:** Tenagework Eseyneh Dagnaw, Amare Mebrat Delie, Tadele Derbew Kassie, Sileshi Berihun, Hiwot Tesfa.

**Visualization:** Tenagework Eseyneh Dagnaw, Amare Mebrat Delie, Tadele Derbew Kassie, Sileshi Berihun, Hiwot Tesfa.

**Writing – original draft:** Tenagework Eseyneh Dagnaw, Amare Mebrat Delie, Tadele Derbew Kassie.

**Writing – review & editing:** Tenagework Eseyneh Dagnaw, Sileshi Berihun, Hiwot Tesfa, Amare Zewdie.

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
