## [Decision Letter · Decision Letter 0]

2 Aug 2024

PONE-D-24-12869Knowledge, attitude, and practice towards COVID-19 and predictors among students in Ethiopia : Systematic review and meta-analysisPLOS ONE

Dear Dr. Dagnaw,

Thank you for submitting your manuscript to PLOS ONE. After careful consideration, we feel that it has merit but does not fully meet PLOS ONE’s publication criteria as it currently stands. Therefore, we invite you to submit a revised version of the manuscript that addresses the points raised during the review process.

**ACADEMIC EDITOR COMMENTS: ** In Tables 2 to 6: Please remove all vertical lines and retain only three horizontal lines: one under the title, one above the column heading, and one between the column headings and the body of the tablePlease revise your references to make sure that they follow the PLOS ONE referencing styleIt is encouraged to have your manuscript undergo a professional English language edition to suit publication standards 

We look forward to receiving your revised manuscript.

Kind regards,

Festo Casmir Shayo, M.D, M.Med., PhD.

Academic Editor

PLOS ONE

3. Please include a new copy of Table 1 in your manuscript; the current table is difficult to read. Please follow the link for more information: https://blogs.plos.org/plos/2019/06/looking-good-tips-for-creating-your-plos-figures-graphics/

Reviewers' comments:

Reviewer's Responses to Questions

**Comments to the Author**

1. Is the manuscript technically sound, and do the data support the conclusions?

Reviewer #1: Partly

2. Has the statistical analysis been performed appropriately and rigorously? 

Reviewer #1: Yes

3. Have the authors made all data underlying the findings in their manuscript fully available?

Reviewer #1: Yes

4. Is the manuscript presented in an intelligible fashion and written in standard English?

Reviewer #1: No

5. Review Comments to the Author

Reviewer #1: L1: The title can be improved by adding “associated factors” instead of “predictors” given the cross-sectional nature of included studies instead of cohort studies.

Abstract

L23-24: Add “s” to “pooled proportion”; “positive” before “attitude” and “prevention” before “practice”

L32: Add “s” to “pooled proportion”; “positive” before “attitude” and “prevention” before “practice”

Introduction

L71: Add a preposition “in” after “results”

L74: Add “s” to “pooled proportion”

Methods

L83: Add “an” before amendment; “made” instead of “done”; and “an” before additional outcome.

L93-94: Rewrite the sentence “In addition, articles included were cross-sectional study design written only in English until December 10, 2023.” to improve grammar.

Discussion

L236: The term “in the world” after the word “pandemic” is redundant. Please remove it.

L237-238: Change the order of words in “a lot of people have died and suffered from it”: suffer to come before death.

L238-239: “The World Health Organization (WHO) does not … or the invention of the drug” is not clear. Please rewrite to improve clarity.

L239-240: The sentence “The best management for the disease is relieving the symptoms, not the virus” needs to be rewritten.

L241-243: The sentence “Regular studies are needed to perform any intervention, advocacy, and health education and communication program, even to change knowledge, attitude, and practice regarding COVID-19 among students in Ethiopia” needs to be rewritten.

L243: The sentence “Due to scarcity among students . . .” is not clear. Please rewrite.

Strengths and limitations

The included studies assessed adequate knowledge, positive attitude, and prevention practice by different sets of questions formulated by respective authors and not standardized. This variability in the assessment criteria affects pooling. I suggest this to be considered as a limitation.

L318: Use the word “generalizability” instead of “generalization”.

Conclusions

L327-328: Rewrite the sentence “The pooled proportion of attitudes and practices eradicated in 2021 from 2020 year indicates the need for more intervention, advocacy, and policy change”

L329: Add “positive” before “attitudes”.

L330: “The result also pointed out the need to give attention for male” should be rewritten as “attention to” “females” instead of “males” as per your results.

References

Many of the references need to be rewritten.

Fig 3 and 6 headings: add “positive” before “attitude”.

6. PLOS authors have the option to publish the peer review history of their article (what does this mean?). If published, this will include your full peer review and any attached files.

Reviewer #1: **Yes: **Dr. Basil Tumaini

---

## [Author Response · Author response to Decision Letter 0]

14 Oct 2024

RESPONSE TO REVIEWERS

Submission ID: PONE-D-24-12869

Title: Knowledge, attitude, and practice towards COVID-19 and associated factors among students in Ethiopia: Systematic review and meta-analysis

Journal Name: PLOS ONE

Academic Editor: Festo Casmir Shayo, M.D, M.Med., PhD

Date: 10/10/2024

Dear editors and reviewers

Thank you for providing valuable comments, suggestions, and coordination to improve the quality of our manuscript. As per your request, we have made appropriate revisions, which are highlighted in green in the 'Revised Manuscript with Track Changes' document. The original document titled 'manuscript' has been submitted separately. We have also provided a point-by-point response to each reviewer's and editor's comments, with page numbers referring to the revised manuscript file. We are looking forward to the outcome of your assessment. Thank you for your valuable time and consideration. 

Academic editor comments

1. In Tables 2 to 6: Please remove all vertical lines and retain only three horizontal lines: one under the title, one above the column heading, and one between the column headings and the body of the table

Authors’ response: Thank you for your comment, correction was made from Table 2 to 6. 

2. Please revise your references to make sure that they follow the PLOS ONE referencing style

Authors’ response: Thank you for your comment. The comment is correct, and the necessary correction has been made.

3. It is encouraged to have your manuscript undergo a professional English language edition to suit publication standards

Authors’ response: Thank you for your comment. The comment is correct, and the necessary correction has been made.

Reviewers' comments:

Reviewer's Responses to Questions

Comments to the Author

1. Is the manuscript technically sound, and do the data support the conclusions?

Reviewer #1: Partly

Authors’ response: Thank you for your comment, we corrected have the conclusion part.

2. Has the statistical analysis been performed appropriately and rigorously?

Reviewer #1: Yes

Authors’ response: Thank you for your comment. 

3. Have the authors made all data underlying the findings in their manuscript fully available?

Reviewer #1: Yes

Authors’ response: Thank you for your comment. 

4. Is the manuscript presented in an intelligible fashion and written in standard English?

Reviewer #1: No

Authors’ response: Thank you for your comment. The comment is correct, and the necessary correction has been made.

5. Review Comments to the Author

1. Reviewer #1: L1: The title can be improved by adding “associated factors” instead of “predictors” given the cross-sectional nature of included studies instead of cohort studies.

Authors’ response: Thanks for your comment. The reviewer was right, and the change was made on page 1.

Abstract

2. L23-24: Add “s” to “pooled proportion”; “positive” before “attitude” and “prevention” before “practice”

Authors’ response: Thanks for your comment, the reviewer was right, and the change was made on page 2.

3. L32: Add “s” to “pooled proportion”; “positive” before “attitude” and “prevention” before “practice”

Authors’ response: Thanks for your comment, the reviewer is right, and the change was made on page 2.

Introduction

4. L71: Add a preposition “in” after “results”

Authors’ response: Thanks for your comment, the reviewer is right, and the change was made on page 4.

5. L74: Add “s” to “pooled proportion”

Authors’ response: Thanks for your comment, the reviewer is right, and the change was made on page 4.

Methods

6. L83: Add “an” before amendment; “made” instead of “done”; and “an” before additional outcome.

Authors’ response: Thanks for your comment, the reviewer is right, and the change was made on page 5.

7. L93-94: Rewrite the sentence “In addition, articles included were cross-sectional study design written only in English until December 10, 2023.” to improve grammar.

Authors’ response: Thanks for your comment, the reviewer is right, and the change was made on page 5.

Discussion

8. L236: The term “in the world” after the word “pandemic” is redundant. Please remove it.

Authors’ response: Thanks for your comment; correction was made on page 17.

9. L237-238: Change the order of words in “a lot of people have died and suffered from it”: suffer to come before death.

Authors’ response: Thanks, the reviewer is right and the correction was made on page 17.

10. L238-239: “The World Health Organization (WHO) does not … or the invention of the drug” is not clear. Please rewrite to improve clarity.

Authors’ response: Thanks for your comment; correction was made on page 17.

11. L239-240: The sentence “The best management for the disease is relieving the symptoms, not the virus” needs to be rewritten.

Authors’ response: Thanks for your comment; correction was made on page 17.

12. L241-243: The sentence “Regular studies are needed to perform any intervention, advocacy, and health education and communication program, even to change knowledge, attitude, and practice regarding COVID-19 among students in Ethiopia” needs to be rewritten.

Authors’ response: Thanks for your comment; correction was made on page 17.

13. L243: The sentence “Due to scarcity among students . . .” is not clear. Please rewrite.

Authors’ response: Thanks for your comment; correction was made on page 17.

Strengths and limitations

14. The included studies assessed adequate knowledge, positive attitude, and prevention practice by different sets of questions formulated by respective authors and not standardized. This variability in the assessment criteria affects pooling. I suggest this to be considered as a limitation.

Authors’ response: Thanks for your comment, the reviewer is right. Correction was made on page 21.

15. L318: Use the word “generalizability” instead of “generalization”.

Authors’ response: Thanks for your comment; Correction was made on page 21.

Conclusions

16. L327-328: Rewrite the sentence “The pooled proportion of attitudes and practices eradicated in 2021 from 2020 year indicates the need for more intervention, advocacy, and policy change”

Authors’ response: Thanks for your comment; Correction was made on page 21.

17. L329: Add “positive” before “attitudes”.

Authors’ response: Thanks for your comment; Correction was made on page 21.

18. L330: “The result also pointed out the need to give attention for male” should be rewritten as “attention to” “females” instead of “males” as per your results.

Authors’ response: Thanks for your comment; Correction was made on page 21.

References

19. Many of the references need to be rewritten.

Authors’ response: Thanks for your comment, the reviewer is right, and the change was made.

20. Fig 3 and 6 headings: add “positive” before “attitude”.

Authors’ response: Thanks for your comment, the reviewer is right and the correction was made on Fig 3 and Fig 6.

---

## [Decision Letter · Decision Letter 1]

12 Nov 2024

Knowledge, attitude, and practice towards COVID-19 and associated factors among students in Ethiopia : Systematic review and meta-analysis

PONE-D-24-12869R1

Dear Dr. Tenagnework Eseyneh Dagnaw,

We’re pleased to inform you that your manuscript has been judged scientifically suitable for publication and will be formally accepted for publication once it meets all outstanding technical requirements.

Kind regards,

Festo Casmir Shayo, M.D, M.MMED, PhD.

Academic Editor

PLOS ONE

Additional Editor Comments (optional):

Reviewers' comments:

Reviewer's Responses to Questions

**Comments to the Author**

1. If the authors have adequately addressed your comments raised in a previous round of review and you feel that this manuscript is now acceptable for publication, you may indicate that here to bypass the “Comments to the Author” section, enter your conflict of interest statement in the “Confidential to Editor” section, and submit your "Accept" recommendation.

Reviewer #1: All comments have been addressed

2. Is the manuscript technically sound, and do the data support the conclusions?

Reviewer #1: Yes

3. Has the statistical analysis been performed appropriately and rigorously? 

Reviewer #1: Yes

4. Have the authors made all data underlying the findings in their manuscript fully available?

Reviewer #1: Yes

5. Is the manuscript presented in an intelligible fashion and written in standard English?

Reviewer #1: Yes

6. Review Comments to the Author

Reviewer #1: The authors have responded to review comments satisfactorily.

The manuscript may thus be considered for publication in PLOS ONE.

7. PLOS authors have the option to publish the peer review history of their article (what does this mean?). If published, this will include your full peer review and any attached files.

Reviewer #1: **Yes: **Dr. Basil Tumaini

---

## [Editor Report · Acceptance letter]

28 Nov 2024

PONE-D-24-12869R1 

PLOS ONE

Dear Dr. Dagnaw, 

I'm pleased to inform you that your manuscript has been deemed suitable for publication in PLOS ONE. Congratulations! Your manuscript is now being handed over to our production team.

Kind regards, 

on behalf of

Dr. Festo Casmir Shayo 

Academic Editor

PLOS ONE